# Hexavalent Chromium Removal from Water and Wastewaters by Electrochemical Processes: Review

**DOI:** 10.3390/molecules28052411

**Published:** 2023-03-06

**Authors:** Işık Kabdaşlı, Olcay Tünay

**Affiliations:** Environmental Engineering Department, Civil Engineering Faculty, İstanbul Technical University, Ayazağa Campus, Istanbul 34469, Turkey

**Keywords:** chemical reduction, electrocoagulation, hexavalent chromium, operating parameters, removal mechanism, sacrificed electrodes, redox electrodes, dimensionally stable electrodes

## Abstract

Hexavalent chromium (Cr(VI)) is a toxic, mutagenic, teratogenic, and carcinogenic species. Its origin is in industrial activities. Therefore, its effective control is realized on a source basis. Although chemical methods proved effective in removing Cr(VI) from wastewaters, more economic solutions with a minimum sludge production have been sought. Among them, the use of electrochemical processes has emerged as a viable solution to the problem. Much research was conducted in this area. The aim of this review paper is to make a critical evaluation of the literature on Cr(VI) removal by electrochemical methods, particularly electrocoagulation with sacrificial electrodes, and to assess the present data as well as to point out the areas that need further elaboration. Following the review of the theoretical concepts of electrochemical processes, the literature on the electrochemical removal of Cr(VI) was evaluated on the basis of important elements of the system. Among them are initial pH, initial Cr(VI) concentration, current density, type and concentration of supporting electrolyte, and the material of electrodes and their operating characteristics and process kinetics. Dimensionally stable electrodes that realize the reduction process without producing any sludge were evaluated separately. Applications of electrochemical methods to a wide spectrum of industrial effluents were also assessed.

## 1. Introduction

Chromium is a transient metal that is relatively abundant in the earth’s crust. Oxidation states of chromium vary from +6 to +2. The most stable and, therefore, dominant forms are Cr(VI) and Cr(III) [1,2]. Cr(III) is common in the soil as a mineral, in ultramafic igneous and metamorphic rocks. Cr(III) is stable and has a strong affinity for particle surfaces. Cr(III) hydroxide is a sparingly soluble salt; therefore, the mobility of Cr(III) is limited. Cr(III) oxidation can only be realized in the soil by manganese oxides. Cr(III) is much less toxic than Cr(VI) and it is a vital element for life, being used for lipid, amino acid, and glycose digestion [2,3,4,5,6]. The majority of Cr(VI) in the environment is from anthropogenic sources. A wide spectrum of industrial activities involves the use of chromium, among them, are electroplating, leather tanning, mineral processing, metallurgical processes, mining, paints, pigment, and glass [2,3,6,7,8]. Cr(VI) is an oxidant, it is reactive and mobile since it is not adsorbed in most sediment, particularly after pH 7. Cr(VI) is highly toxic, mutagenic teratogenic, and carcinogenic. It causes many health effects on humans such as renal impairment, neutral cell injury, liver dysfunction, and stomach ulcers [7,9,10]. Chromium chemistry in soils is well studied. Cr(VI) has an affinity to water, is not sorbed in most sediments, and migrates above neutral pH. Organic matters, particularly those having sulfhydryl groups and humic and fulvic acids, reduce Cr(VI). It is also reduced by ferrous iron (Fe^2+^) and S^2−^ [1,6,7,9,10].

The above information indicates the need for control and treatment of chromium–laden wastewaters at the source. There are various techniques used for the treatment of chromium in wastewaters. Among them are ion exchange, adsorption using a wide spectrum of natural and waste material, membrane processes such as electrodialysis, electro-ionization used separately or coupled with ion exchange, and biological treatment using bacteria, fungi, yeast, or algae, all suffering from the production of high amounts of contaminated waste material, sludge, or concentrate [4,7,11]. There are also less common methods such as photocatalysis and solvent extraction [12,13]. On the other hand, the chemical reduction of Cr(VI) followed by the chemical precipitation of Cr(III) has some drawbacks such as the requirement of higher doses of chemicals for diluted wastewaters and excess sludge production, which are still being used as a reliable method [14]. The alternative came from the idea of producing the reductant mainly Fe(II) in situ by electrolysis. Electrochemical methods have begun to be used or tested for several decades for chromium removal from wastewaters and proved to be efficient and flexible methods. Therefore, these methods are considered the most promising technology on which substantive research has been made.

Electrochemical methods named electroreduction and electrocoagulation, or shorthand notation as EC methods, combine chromium reduction and precipitation in a single step. The process is based on electrolysis. While solubilized anode material such as Fe^2+^ or Al^3+^ carries out chromium reduction, increasing pH through the conversion of hydrogen ions to hydrogen gas at the cathode precipitate Cr(III) as Cr(OH)_3_. The studies on the subject are reflected by scientific publications as well as review papers indicating increasing improvements and new applications in the field [15,16]. The material of electrodes in the EC process has been the subject of a number of studies. While metallic electrodes such as iron, aluminium, and copper were widely used, non-sacrificial electrodes made of carbon, gold, and conducting polymers were also tested for the performance [15,17,18]. Carbon electrodes as carbon felt and graphite felt have been utilized as cathode [16]. Electrode shape and area, reactor types such as the use of column reactor, and the use of direct and pulse current were tested as potential applications [3,19,20]. The optimization of basic operation parameters of EC such as initial pH, electrode distancing, current density, electrolyte concentration, and reaction time was the subject of a significant number of studies where iron electrodes were preferentially used [8,20,21,22,23,24]. Several studies dealt with the kinetics of the EC process as well as pH adjustment using real-time control [8,20,25]. There are a few studies that attempted to make an economic analysis of the EC process [20,26,27].

This paper aims to review and evaluate the latest developments in the area of hexavalent chromium removal by EC process applications. The literature data were analysed to assess the critical points in the operation such as initial pH, current density, supporting electrolyte concentration, and initial Cr(VI) concentration, which are the determining factors of the economics of the processes, rate of the process, and the most important the efficiency of the process. New developments in the field were also taken into consideration. In this context, non-sacrificial electrodes were critically evaluated since they are a step ahead of minimizing sludge production, which is the main advantage of the EC process over the chemical reduction and precipitation process.

## 2. Responsible Chromium Removal Mechanisms in the Electrocoagulation Process

In the electrocoagulation process, Cr(VI) can be reduced to Cr(III) (i) during reactions, taking place at the cathode or (ii) at the anode surface under acidic conditions. Then, Cr(III) is precipitated or co-precipitated with metal hydroxides as Cr(OH)_3_. Other possible removal mechanisms are (i) specific adsorption of Cr (VI) by direct complexation and surface complexation reactions with soluble and insoluble metal hydroxides (or oxyhydroxides), respectively [28], and (ii) adsorption through electrical neutralization and electrostatic attraction between Cr(VI) and charged metal hydroxides [28]. One or more of these mechanisms are responsible for chromium removal using the electrocoagulation process depending on the material of the sacrificed electrode, i.e., iron or aluminium. Therefore, chromium removal mechanisms will be introduced for each electrode material separately in the following subsections.

### 2.1. Iron-Based Electrodes

The main reactions taking place at the electrode surface during electrocoagulation using iron or steel electrodes are as follows [25,29]:(1)$${\mathrm{Fe}}_{\left(s\right)}-{\;2e}^{-}\;\to {\;\mathrm{Fe}}_{\left(\mathrm{aq}\right)}^{2+}\;E^{o}=-0.447\;V\;\mathrm{at}\;\mathrm{the}\;\mathrm{anode}$$
(2)$$2H_{2}{O+2e}^{-}\;\to {\;H}_{2}{}_{\left(g\right)}{+2\mathrm{OH}}^{-}\;E^{o}=-0.828\;V\;\mathrm{at}\;\mathrm{the}\;\mathrm{cathode}$$

In the case of oxygen formation at the anode, further oxidation of Fe^2+^ to Fe^3+^ can be realized according to the following reaction [19,30]:4Fe^2+^ + O_2_ + 2H_2_O → 4Fe^3+^ + 4OH^−^ → in alkaline conditions(3)
(4)$${4\mathrm{Fe}}^{2+}{+4H}^{+}{+O}_{2}\to {\;4\mathrm{Fe}}^{3+}{+2H}_{2}O\;\to \mathrm{in}\;\mathrm{acidic}\;\mathrm{conditions}$$

Overall electro-dissolution reaction can be obtained by a combination of Equations (1) and (4) as follows:Fe_(s)_ + 2H_2_O → {Fe(OH)_2_} + H_2(g)_(5)

{Fe(OH)_2_} represents the products (Fe(OH)_2(s)_; FeOH^+^ + OH^−^; Fe^2+^ + 2OH^−^) generated during electro-dissolution process in bulk solution depending on pH [25]. Under the acidic condition, iron can also chemically dissolve:Fe_(s)_ + 2H^+^ → Fe^2+^ + H_2(g)_(6)

As aforementioned, Cr(VI) can be reduced to Cr(III) by several mechanisms during electrocoagulation with iron-based electrodes. The reduction by Fe^2+^ released by electro-dissolution (Equation (4)) or chemical dissolution (Equation (5)) is the predominant mechanism for chromium removal [3,19,23,25,30,31,32,33]:

At pH < 6.5:(7)$${\mathrm{HCrO}}_{4}^{-}{+3\mathrm{Fe}}^{2+}{+7H}^{+}\to {\;\mathrm{Cr}}^{3+}{+3\mathrm{Fe}}^{3+}{+4H}_{2}O$$
(8)$${\mathrm{Cr}}_{2}O_{7}{}^{2-}{+6\mathrm{Fe}}^{2+}{+14H}^{+}\to {\;2\mathrm{Cr}}^{3+}{+6\mathrm{Fe}}^{3+}{+7H}_{2}O$$

At 6.5 < pH < 7.5:(9)$${\mathrm{CrO}}_{4}{}^{2-}{+3\mathrm{Fe}}^{2+}{+4H}_{2}O\to {3\mathrm{Fe}}^{3+}{+\mathrm{Cr}}^{3+}{+8\mathrm{OH}}^{-}$$

At pH > 7.5:(10)$${\mathrm{CrO}}_{4}{}^{2-}+3\mathrm{Fe}{\left(\mathrm{OH}\right)}_{2}{+4H}_{2}O\;\to \;\mathrm{Cr}{\left(\mathrm{OH}\right)}_{3}{}_{\left(s\right)}+3\mathrm{Fe}{\left(\mathrm{OH}\right)}_{3}{}_{\left(s\right)}{+2\mathrm{OH}}^{-}$$

Electrochemical reduction of Cr(VI) may also occur at the cathode [25,30,34,35]:(11)$${2\mathrm{HCrO}}_{4}{}^{-}{+3\mathrm{Fe}+4H}_{2}{O+2H}^{+}\to \;2\left\{{\mathrm{Cr}(\mathrm{OH})}_{3}\right\}+3\left\{{\mathrm{Fe}(\mathrm{OH})}_{2}\right\}$$
(12)$${\mathrm{Cr}}_{2}O_{7}{}^{2-}{+3\mathrm{Fe}+5H}_{2}{O+2H}^{+}\to \;2\left\{{\mathrm{Cr}(\mathrm{OH})}_{3}\right\}+3\left\{{\mathrm{Fe}(\mathrm{OH})}_{2}\right\}$$
(13)$${\;2\mathrm{CrO}}_{4}{}^{-}{+3\mathrm{Fe}+4H}_{2}{O+4H}^{+}\to \;2\left\{{\mathrm{Cr}(\mathrm{OH})}_{3}\right\}+3\left\{{\mathrm{Fe}(\mathrm{OH})}_{2}\right\}$$
where {Cr(OH)_3_} and {Fe(OH)_3_} are the species produced during electrocoagulation Cr(OH)_3(s)_; ${\mathrm{Cr}(\mathrm{OH})}_{4}^{-}{+H}^{+}$;${\;\mathrm{Cr}}_{3}{\left(\mathrm{OH}\right)}_{4}^{5+}{+5\mathrm{OH}}^{-}$; Fe(OH)_3(s)_; ${\mathrm{Fe}(\mathrm{OH})}_{2}^{-}{+H}^{+}$; or Fe^3+^ + 3OH^−^ depending on solution pH [25]. The reduction of Cr(VI) by zero-valent iron at the anode surface is another possible chromium removal mechanism in acidic conditions [25,34]:(14)$${2\mathrm{HCrO}}_{4}{}^{-}{+2\mathrm{Fe}+4H}_{2}{O+2H}^{+}\to \;2\left\{{\mathrm{Cr}(\mathrm{OH})}_{3}\right\}+2\left\{{\mathrm{Fe}(\mathrm{OH})}_{3}\right\}$$
(15)$${\mathrm{Cr}}_{2}O_{7}{}^{2-}{+2\mathrm{Fe}+5H}_{2}{O+2H}^{+}\to \;2\left\{{\mathrm{Cr}(\mathrm{OH})}_{3}\right\}+2\left\{{\mathrm{Fe}(\mathrm{OH})}_{3}\right\}$$
(16)$$2{\mathrm{CrO}}_{4}{}^{-}{+2\mathrm{Fe}+4H}_{2}{O+4H}^{+}\to \;2\left\{{\mathrm{Cr}(\mathrm{OH})}_{3}\right\}+2\left\{{\mathrm{Fe}(\mathrm{OH})}_{3}\right\}$$

Depending on reaction pH, Fe^3+^ also hydrolyses to form soluble monomeric, dimeric, and possibly small polymeric hydroxo metal complexes such as Fe(OH)_3_$;\;\mathrm{Fe}{\left(\mathrm{OH}\right)}^{2+}$; $\mathrm{Fe}{\left(\mathrm{OH}\right)}_{2}^{+}$ $\mathrm{Fe}{\left(\mathrm{OH}\right)}_{4}^{-}$;$\;{\mathrm{Fe}(H}_{2}{O)}_{3}{\left(\mathrm{OH}\right)}_{3}$; $\mathrm{Fe}{\left(H_{2}O\right)}_{6}^{3+}$; $\mathrm{Fe}{(H_{2}O)}_{4}{\left(\mathrm{OH}\right)}^{+}$; $\mathrm{Fe}{(H_{2}O)}_{5}{\left(\mathrm{OH}\right)}^{2+}$; ${\mathrm{Fe}}_{2}{{(H}_{2}O)}_{8}{\left(\mathrm{OH}\right)}_{2}^{4+}$; and ${\mathrm{Fe}}_{2}{{(H}_{2}O)}_{6}$ [3,19,23,36]. These hydroxo complex species can remove negatively charged chromium species either by complexation or electrostatic attraction [3,23,28,36]. Another removal mechanism is also co-precipitation of Cr(III) and Fe(III) as Cr*_x_*Fe_1−*x*_(OH)_3_ at pH 2–4 as a consequence of hydroxyl ion production due to the reaction at the cathode [23,37]. Figure 1 illustrates all the above-mentioned possible chromium removal mechanisms schematically.

### 2.2. Aluminium Electrodes

For aluminium material, the main electrochemical reactions taking place at electrode surfaces during electrocoagulation process are as follows [31]:Al_(s)_ – 3e^−^ → Al^3+^_(aq)_ (E^o^= −1.669 V) at the anode(17)
2H_2_O + 2e^−^ → H_2_ + 2OH^−^ at the cathode(18)

Al^3+^ ions generated by electro-dissolution of the anode (Equation (17)) cannot take part in the oxidation-reduction reactions and act only as a coagulant [31]. Therefore, it is assumed that the electrochemical reduction of Cr(VI) to Cr(III) occurs at the cathode depending on the solution pH:(19)$${\mathrm{Cr}}_{2}O_{7}{}^{2}{+6e}^{}{+14H}^{+}\to {\;2\mathrm{Cr}}^{3+}{+7H}_{2}O\;\mathrm{in}\;\mathrm{acidic}\;\mathrm{medium}$$
(20)$${\mathrm{CrO}}_{4}{}^{2-}{+3e^{-}}^{}{+4H}_{2}O\;\to {\;\mathrm{Cr}}^{3+}{+8\mathrm{OH}}^{-}\;\mathrm{in}\;\mathrm{alkaline}\;\mathrm{medium}$$

Meanwhile the electro-generated Al^3+^ ions hydrolyse to form various monomeric and polymeric aluminium hydroxo-complexes. While the anode vicinity becomes acidic due to these hydrolysis reactions, the significant increase of the local pH at the cathode vicinity as the consequence of hydrogen evolution (Equation (18)) induces a chemical attack of aluminium and its hydroxide film according to the following reactions [31,38]:(21)$${2\mathrm{Al}}_{\left(s\right)}{+6H}_{2}{O+{\mathrm{OH}}^{-}}^{}\to 2\mathrm{Al}{(\mathrm{OH}{)}^{-}}_{4\;}^{}{+3H}_{2(g)}\;$$
(22)$${\mathrm{Al}(\mathrm{OH})}_{3}+{\mathrm{OH}}^{-}\to {\mathrm{Al}(\mathrm{OH})}_{4}^{-}$$

The $\mathrm{Al}(\mathrm{OH}{)}_{4}^{-}$ formed at the cathode vicinity is converted into amorphous Al(OH)_3_ in the bulk solution. Furthermore, when the solution pH shifts towards neutral or slightly alkali values owing to the production of hydroxyl ions at the cathode, Cr(OH)_3_ precipitates. All the above-mentioned possible chromium removal mechanisms are depicted schematically in Figure 2.

## 3. Operating Parameters

Initial pH and solution pH attained at the end of the electrocoagulation operation play a determining role on the effluent quality and sludge characteristics as well as chromium removal efficiency. Applied current or current density, initial Cr(VI) concentration, type and dose of electrolyte, and material of anode and cathode also have a great influence on the process performance as well as operation cost and sludge production. The effect of all mentioned operating parameters will be discussed in the following subsections for each electrode material.

### 3.1. Iron-Based Electrodes

#### 3.1.1. Initial pH

Initial pH is a key parameter determining both the rate of the reduction of Cr(VI) to Cr(III) and total chromium ((Cr(VI) + Cr(III)) removal efficiency. The reduction reaction strictly depends on the initial pH as well as the solution pH reached at any time due to continuous hydroxyl ion production (Equation (2) [28]. Acidic pH conditions accelerate the production of Fe^2+^ acting as a reducing agent by electrolytic oxidation of the anode [25,28,35,39]. Extremely acidic initial pH conditions may lead to effluents of acidic character at which both Cr^3+^ and Fe^3+^ remain in the effluent without precipitating as seen in Equations (7) and (8) [23,25,40]. Such high solubilities were explained by the formation of their positively charged soluble hydroxo-complex species. For instance, CrOH^2+^ is the dominant chromium species pH from 3.8 to 6.4 [41], and FeOH^2+^ and F${e(\mathrm{OH})}_{2}^{+}$ are the dominant species at pH values below 5 [42]. In alkali pH values, negatively charged soluble hydroxo-complex species become dominant, resulting in high remaining Cr(III) and Fe^3+^ concentrations [40]. Moreover, high concentrations of hydroxyl ions trigger the formation of small and dense Fe(OH)_3_ flocs, which are difficult to be attached by the gas bubbles. This difficulty creates sludge floatation and a separation problem [40]. Considering that residual chromium and iron concentrations are extremely high at pH values below 3 and over 10, some researchers reported that neutral initial pH values such as 4–8 [43], 6–8 [40], and 6 [3,28] as optimum pH or pH range. Conversely, in some studies, optimum pH was determined as extremely acidic values such as 1 [19,44], 2 [39], 3 [45], 4 [46,47], 4.5 [48], and 4.9 [34]. These data suggest that solution pH at any time and at the end of electrocoagulation rather than the initial pH is of importance. If enough hydroxyl ions are produced according to Equation (2), the buffer capacity of the solution will be exceeded, and the initial acidic pH shifts towards an alkaline region where precipitate and/or co-precipitation of Cr(III)/Fe^3+^ are possible. Such an electrocoagulation application makes it possible to achieve the maximum chromium removal performance.

pH evolution during electrocoagulation is affected by some operation parameters. Xu et al. [25] investigated the effect of initial Cr(VI) concentration (10, 52, and 94 mg/L) and initial pH (pHi = 2, 3, and 4)) on the pH evolution and found a relation between p[Cr(VI)] and initial pH. In the case of p[Cr(VI)] = pHi, the final pH values were alkaline; when p[Cr(VI)] was smaller or greater than pHi, it reached alkali and acidic values, respectively. Based on the rate of Cr(VI) removal, four stages were described, namely (i) rapid removal, (ii) constant removal, (iii) decelerating removal, and (iv) complete removal [25]. When p[Cr(VI)] = pHi, in the first stage, the combined effect of reductions of chemical, cathodic, and zero-valent iron yielded a rapid Cr(VI) removal. In the second stage, ferrous iron dissolution was found to be a rate-limiting step with a constant Cr(VI) reduction rate. During the third stage, a reduction in DO in the reaction solution and a significant increase in the pH dramatically decelerated the Cr(VI) removal rate. In the last stage, the solution pH remained constant at around pH 7 during the simultaneous oxidation of ferrous iron and the formation of Fe(OH)_3_. When p[Cr(VI)] < pH_i_, the third stage occurred before the second stage as a result of a rapid Fe^2+^ oxidation with DO with an increasing pH to 7.0 at the end of the first stage. In the case of p[Cr(VI)] > pH_i_, the stages of II and III disappeared due to faster Cr(VI) reduction in acidic conditions, and high residual chromium concentration was measured at the end of electrocoagulation (final pH < 4.5) [25].

#### 3.1.2. Initial Cr(VI) Concentration

High remaining Cr(VI) concentrations in effluents were also obtained at extremely high initial chromium concentrations. Scientific data indicated that elevating initial chromium concentrations resulted in an increase in remaining chromium concentrations for a constant current density [3,19,20,39,46,48,49]. In the study of Das and Nandi [46], the residual chromium concentration increased from 0.0046 to 44.205 mg/L with an increase in initial Cr(VI) concentration from 10 to 100 mg/L at the end of a 60 min electrocoagulation operation at 43.03 A/m^2^. Similarly, El-Taweel et al. [19] reported that an increase in the initial Cr(VI) concentration from 40 to 200 mg/L caused an increase in the residual Cr(VI) concentration from 0 to 107 mg/L. In all studies, this behaviour was explained by a constant amount of Fe^2+^ being released due to electro-dissolution of the anode at a constant current density which was insufficient to reduce all of Cr(VI) ions [3,19,39,46,49]. Therefore, a proportional increase either in current density or in electrode surface area are recommended for enough Fe^2+^ production to minimize residual Cr(VI) in the effluent [42]. Such unsatisfactory removal efficiencies were also attributed to the increase in the amount of Cr(VI) adsorbed onto the anode surface resulting in the anode passivation [43].

#### 3.1.3. Current Density or Current

It is well known that an increase in current density improves the metal removal efficiency [19,22,31,39,43,45,46,47,50,51]. Current density determines the rate of electrochemical metal dosing to the water, the rate and size of electrolytic bubble production, and the flocs growth [47,52,53]. Bubble density increases while its size decreases with increasing current density, bringing about a greater upwards flux and a faster removal of pollutants. Current density directly affects the electrical energy requirement and electrode material consumption, and, consequently, the operating cost of the electrocoagulation process [28]. In the study of Lu et al. [28], increasing current density from 0.42 to 0.94 mA/cm^2^ enhanced the Cr(VI) removal efficiency from 74.35 to 100% while increasing the energy consumption from 0.24 to 0.94 kWh/m^3^. The complete chromium removal was attained at the highest energy consumption for an initial Cr(VI) concentration of 106 mg/L, at an initial pH of 6, and for an electrolysis time of 50 min. El-Taweel et al. [19] also reported that increasing the current from 0.2 to 1 A caused an increase in energy consumption from 0.002 to 0.009 kWh per gram of Cr(VI) removed, and in iron consumption from 0.02 to 0.37 g iron per gram of Cr(VI) removed for an initial Cr(VI) concentration of 140 mg/L, initial pH of 4.66, an NaCl concentration of 1 g/L, and an electrolysis time of 14 min. Similarly, an extension in electrolysis time led to an increase in energy consumption together with an improvement in Cr(VI) removal efficiency [20,54].

According to Heidmann and Calmano [30], two different Cr(VI) removal mechanisms were possible at high currents (1.0–3.0 A) and at low currents (0.05–1.0 A) for an initial Cr(VI) concentration of 20 mg/L and initial pH 5–6. Their data obtained at high currents indicated that Cr(VI) removal efficiency lines were nearly congruent, an increase in current did not accelerate the process, and the process seemed to be mainly controlled by the chromium concentration. Therefore, they concluded that Cr(VI) was directly reduced at the cathode. At low currents, an increasing current accelerated the chromium removal due to an increase in the production of both Fe^2+^ and OH^−^. Based on this improvement, the chemical reduction by Fe^2+^ dissolved from electrodes was assumed to be the dominant removal mechanism. 

#### 3.1.4. Supporting Electrolyte

The energy consumption of the electrochemical process is reduced by the addition of an electrolyte. The addition of a supporting electrolyte improves the Cr(VI) removal efficiency since the conductivity of the reaction solution increases and the anode passivation is prevented [23]. Scientific data on this subject confirmed higher or almost complete Cr(VI) removal efficiencies within a shorter electrolysis time attained by the addition of the proper amount of electrolyte. Salts of monovalent ions are reported as the best supporting electrolytes [55,56]. Among them, NaCl is the most common electrolyte used in electrocoagulation applications. In addition to NaCl [19,23,32,44,46,48], NaNO_3_ [32,44,46], Na_2_SO_4_ [19,32,44,46], and H_2_SO_4_ [23,50,57] were also tested as supporting electrolytes in these studies. Das and Nandi [58] found that NaCl was superior to NaNO_3_ and Na_2_SO_4_ in a dosage range of 0.33–0.83 g/L, for electrocoagulation performed at an initial pH of 4.0, a current density of 43.103 A/m^2^, and an initial Cr(VI) concentration of 40 mg/L for 60 min electrolysis time. At the maximum electrolyte dose (0.83 g/L), Cr(VI) removal efficiencies were determined as 99.99% for NaCl, 99.94% for NaNO_3,_ and 99.18% for Na_2_SO_4_. They concluded that chloride ions improved the electro-dissolution of the anode by destroying the iron oxide film formed on the anode, while nitrate and sulphate anions interfered with its dissolution, resulting in a lower production of iron flocs in the solution. In their study, the lowest energy consumption was obtained by the addition of 0.83 g/L NaCl corresponding to the highest dose tested. Mouedhen et al. [31] also compared the effect of NaCl (0.5 g/L) and Na_2_SO_4_ (1 g/L) on solution pH change, residual chromium, and iron concentrations. Electrocoagulation operating conditions were the initial Cr(VI) concentration of 45 mg/L, current density of 1 A/dm^2^, and initial pH of 7.4 and 7.2 for Na_2_SO_4_ and NaCl, respectively. pH changes were practically the same for both supporting electrolytes, and the pH rose from an initial value to reach a steady state at 11. Almost complete chromium removal was achieved after 15 min and dissolved iron did not exceed 0.1 mg/L during the electrocoagulation operation for both electrolytes. These data, reported for Na_2_SO_4_ and NaNO_3_ as supporting electrolytes, did not conform to those of Aber et al. [44] and Lakshmipathiraj [32] whose studies indicated that extremely low (<20%) reduction efficiencies were obtained in the presence of both electrolytes.

H_2_SO_4_ was used together with NaCl to create an acidic medium [23]. The experimental study results proved that (i) chloride ions enhanced the anode dissolution by pitting corrosion and favoured the reduction of Cr(VI) to Cr(III) and the subsequent precipitation of Fe^3+^/Cr^3+^ hydroxides, particularly at low concentrations of H_2_SO_4_ (0.001 and 0.01 M); (ii) when higher pH values were attained in some operation conditions such as the higher NaCl concentration (1 g/L) and the lower H_2_SO_4_ concentration (0.001 M), the residual iron concentrations were low. Zewail and Yousef [48] investigated the effect of NaCl concentration (0.5; 1.0 and 2.0 g/L) on the removal efficiency of Cr(VI) and Cr(III) for an initial chromium concentration of 150 mg/L at 10.02 A/m^2^. Electrocoagulation was operated at an initial pH of 3.4 and 4.5 and for a time of 10 and 20 min for Cr(III) and Cr(VI), respectively. As expected, Cr(VI) removal efficiency improved with increasing NaCl concentration. However, Cr(III) removal efficiency decreased beyond 1 g/L NaCl concentration since the solubility of Cr(OH)_3_ increased with increasing ionic strength. A similar negative effect was also reported for NaCl concentrations greater than 0.7 g/L [23] and 1.5 g/L [19].

It should be noted that while an overdose of electrolytes induces overconsumption of the anode due to a pitting corrosion [52], an insufficient dose does not prevent passivation. Therefore, electrolyte addition needs to be optimized. Xu et al. [59] proposed an approach based on galvanostatic measurements to optimize chloride ions for depassivation and to determine an optimum chloride concentration for the pitting dissolution of iron electrodes during electrocoagulation. This optimum chloride concentration was described as a concentration preventing passivation during the evolution of pH and Cr(VI) reduction in the process.

### 3.2. Aluminium Electrodes

In the literature, only a few studies have focused on the Cr(VI) removal by electrocoagulation utilizing aluminium electrodes as both anode and cathode. The published data on operating parameters affecting the process performance will be summarized as follows.

The effect of the initial pH on the Cr(VI) removal efficiencies was investigated in some studies [60,61,62,63,64,65,66]. Rezaee et al. [64] explored the simultaneous removal of Cr(VI) and nitrate ions from an aqueous solution containing both pollutants. The results of electrocoagulation initialized at pH 4, 6, and 8 indicated that Cr(VI) removal efficiency decreased with increasing initial pH, and pH 4 yielded the highest Cr(VI) removal. Similar conclusions were also made by the other authors. Yu et al. [65] reported that the best Cr(VI) removal ratio was obtained at pH 5 as 99.92%. El-Ashtoukty et al. [63] achieved the highest Cr(VI) removal efficiencies within the pH range of 4.5–5.5. Golder et al. [66] performed electrocoagulation at initial pH values of 2, 4.87, 7, and 10, and obtained the highest Cr(VI) removal (42.3%) at pH 4.87. In the study of Kumar and Basu [60], the initial Cr(VI) concentration remained unchanged in the pH range of 4–5; Cr(VI) removal efficiency enhanced with increasing initial pH between 4 and 5 and decreased beyond pH 5.

The above-mentioned data have revealed that the highest Cr(VI) removal efficiencies can be obtained at an initial pH range of 4–5.5 depending on the operating conditions. In fact, the process performance is directly related to the interaction of Al^3+^ with water to form hydrolytic species at any time during the electrocoagulation [66,67]. The formation of these species can be represented as follows [67]:(23)$${\mathrm{xAl}}^{3+}{+\mathrm{yH}}_{2}O\;⇆{\;\mathrm{Al}}_{x}{\left(\mathrm{OH}\right)}_{y}^{3x-y}{+\mathrm{yH}}^{+}$$
where x is changed from 0 to 13 for the polymeric species at intermediate pH values in concentrated aluminium solutions. At extremely acidic and alkaline pH values, the monomeric hydroxo complex species are dominant:(24)$${\mathrm{Al}}^{3+}{+\mathrm{yH}}_{2}O\;⇆{\;\mathrm{Al}}_{x}{\left(\mathrm{OH}\right)}_{y}^{3-y}{+\mathrm{yH}}^{+}$$
where y is changed from 0 to 4. This speciation profoundly affects the solubility of Al(OH)_3_ as a function of pH. The monomeric and polymeric aluminium hydroxo complex species [63,67,68,69] that are dominant with respect to pH are:$${\mathrm{Al}(H}_{2}{O)}_{6}^{3+};\;\mathrm{Al}{\left(H_{2}O\right)}_{5}{\left(\mathrm{OH}\right)}^{2+};{\mathrm{Al}(H}_{2}{O)}_{4}{\left(\mathrm{OH}\right)}_{2}^{+}\;\;\mathrm{pH}\;\text{<}\;4\;{\mathrm{Al}}_{6}{\left(\mathrm{OH}\right)}_{15}^{3+}{\mathrm{Al}}_{8}{\left(\mathrm{OH}\right)}_{20}^{4+}\;\mathrm{pH}\;4\text{–}5$$
Al(OH)_3_(H_2_O)_3_ pH 5.5
$${\mathrm{Al}(\mathrm{OH})}_{4}^{-}\;\mathrm{pH}\;\text{>}\;7$$

The number of positively charged surface sites decreases while pH shifts to pH_zpc_ of Al(OH)_3_ (8.4). In addition to a decrease in active sites to be attached, dissolution of Al(OH)_3_ takes place to form ${\mathrm{Al}(\mathrm{OH})}_{4}^{-}$, having poor coagulative properties at pH > 7 [60,65,66]. Within the pH range of 4–5, polymeric hydroxo complex species with 3 and 4 positive charges are formed; these species attract the negatively charged Cr(VI) anions ($i.e{.,\;\mathrm{CrO}}_{4}^{-})$ and capture them more efficiently than the less charged or negatively charged hydroxo complex species.

Similar to iron electrodes, when aluminium electrodes were used as both anode and cathode, Cr(VI) removal efficiency increased with increasing current density and extending electrolysis time [62,63,64,65,66]. In a study, increasing current density from 10 to 25 mA/cm^2^ resulted in an increase in Cr(VI) removal efficiency from 52.74 to 91.48 % for 60 min, and an extension in electrolysis time from 5 to 60 min brought about a remarkable increase in Cr(VI) removal efficiency from 6.52 to 91.48% [65]. In another study, the highest Cr(VI) removal efficiency (99.99%) was achieved at the highest current density (8.6 A) [62]. This improvement with both current density and electrolysis time was ascribed to (i) increasing the amount of the dissolved Al^3+^ at the anode according to Faraday’s law and (ii) enhancement in mixing conditions with increasing the discharge rate of H_2_ bubbles at the cathode [60,63]. On the other hand, an increase in both current density and electrolysis time caused higher sludge production and specific electrical energy consumption (SEEC) [60,63,64,66,70].

As expected, increased initial Cr(VI) concentration had a negative effect on the Cr(VI) removal efficiency. Cr(VI) removal efficiency drastically reduced at a constant current density and at high initial Cr(VI) concentrations [60,61,65]. This was attributed to the limited in-situ generation of the coagulant which was quickly exhausted from the system and led to a higher residual Cr(VI) in the effluent [46,60,61].

A direct comparison of the systems using iron and aluminium electrodes can hardly be made since there is no work specifically targeted to this purpose. It is even difficult to find experiments conducted at similar conditions, and almost all experiments are bench-scale. Bench-scale experiments do not clearly demonstrate the operation problems nor are they aimed to. While both electrodes can work with high efficiencies, a more precise control of the reduction process seems to be required for aluminium electrodes. The produced aluminium hydroxide flocs serve only to adsorb and entrap chromium. If Cr(VI) cannot be effectively reduced, this may only mean that the Cr(VI) problem is transferred from the water to sludge phase. Aluminium hydroxide sludge is voluminous, and its separation may not be easy. The economic aspects of the systems need to account for sludge handling and disposal. This point will be also emphasized in the applications for industrial effluents section.

## 4. Optimum Operating Conditions and Interactions of Process Variables

Response surface methodology (RSM) is a combination of mathematical and statistical techniques used for developing, improving, and optimizing the processes and used to evaluate the relative significance of several affecting factors even in the presence of complex interactions [71]. This methodology has been employed to evaluate the effects of various operating conditions on the Cr(VI) removal by electrocoagulation with sacrificed electrodes. The process has been optimized using RSM in a combination with the Box-Behnken design (BBD) [24,72,73], Taguchi method [74], and central composite design (CCD) [8,20,21,22,27,57,70] for several process variables such as current density, initial Cr(VI) concentration, initial pH, electrolyte concentration, electrical energy consumption, settled sludge volume, operation cost, reaction temperature, or H_2_SO_4_ dose.

Khan et al. [20] employed a four-factor CCD together with RSM to evaluate the effects of the process parameters on response variables: Cr(VI) removal efficiency and energy consumed per gram removal of chromium for electrocoagulation with a mild steel rod anode and a hollow cylindrical iron mesh cathode. In their study, the optimum conditions were determined as a pH of 3.0, an applied current of 1.48 A, an initial Cr(VI) concentration of 49.96 mg/L, and an electrolysis time of 21.47 min for complete Cr (VI) removal from K_2_Cr_2_O_7_ aqueous solution. For these conditions, energy consumption was found as 12.97 W hour per gram removal of Cr (VI). The average operational cost per gram removal was calculated as 0.116 and 0.084 Indian rupees for electrode material and electricity, respectively. In a study, electrocoagulation with an iron electrode was optimized using CCD with RSM to maximize Cr(VI) removal from forward osmosis reject water and to minimize operating cost, electrical energy consumption, and settled sludge volume [22]. Under optimized conditions (an electrolysis time of 59.7 min and a current of 1.24 A (J = 6.32 mA/cm^2^)), operating costs of 0.014 USD/m^3^, the electrical energy consumption of 0.005 kWh/m^3^, and settled sludge volume of 445 mL/L were obtained for 90.0% chromium removal. Another RSM with CCD modelling study was employed to evaluate the effects and interactions of process variables: applied electric current, electrolyte concentration, and application time on the Cr(VI) removal from a hard chromium coating process effluent (Cr(VI): 1470 mg/L) [21]. For the electrocoagulation with stainless steel, the optimum conditions for complete (100%) Cr(VI) removal were established as 7.4 A applied electric current, 33.6 mM electrolyte (NaCl) concentration, and 70 min application time. Gilhotra et al. [27] applied RSM-based CCD to optimize independent process variables viz. pH, current density, and treatment time. In their optimization study, Cr(VI) removal efficiency and energy were selected as response variables. Under optimized process conditions (a pH of 5, a current density of 68 A/m^2^, and a treatment time of 17 min) 97.5% chromium removal efficiency was achieved by electrocoagulation with SS. Bhatti et al. [70] used RSM with CCD to achieve an energy-efficient removal of Cr(VI) from electrocoagulation with aluminium electrodes. Their data, obtained from predictive models using ANOVA and multiple response optimization, indicated that optimal Cr(VI) removal efficiency (50%) was attained at 11 V and 18.6 min treatment time with a consumption of 15.46 KWh/m^3^ energy.

Patel and Parikh [8] made an optimization using RSM in combination with CDD to investigate the effect of initial Cr(VI) concentration, pH, electrode distance, current density, and supporting electrolyte (NaCl) concentration on Cr(VI) removal from K_2_Cr_2_O_7_ aqueous solution. For the electrocoagulation using copper electrodes, current density of 41.32 A/m^2^, an electrode distance of 1.4 cm, an initial pH of 5.65, an electrolysis time of 40 min, and an initial conductivity of 0.21 ms were determined as optimum operating conditions to achieve 93.33% chromium removal efficiency. RSM with CCD was also employed to evaluate the effect of the H_2_SO_4_ dosage, current intensity, reaction time, and reaction temperature on the chromium removal from K_2_Cr_2_O_7_ aqueous solution by electrocoagulation with a steel electrode [57]. The optimization data indicated that the effect of single factor on Cr(VI) removal efficiency followed the order H_2_SO_4_ dosage > reaction time > reaction temperature > current intensity. Kumar and Basu [60] optimized the removal of Cr(VI) from K_2_Cr_2_O_7_ aqueous solution (with 1 g/L NaCl) by electrocoagulation with vertically rotating cylindrical aluminium electrodes using RSM with CCD. Considering that excess input energy would bring about a marginal improvement in the Cr(VI) removal efficiency, the process that was optimized for maximum Cr(VI) removal corresponded to minimum energy input. Under the optimum conditions (Cr(VI) 34.99 mg/L, 2.189 A, pH_o_ 4.5, and a rotational speed of 72.46 rpm) the Cr(VI) removal efficiency, SEEC, and operating cost were found as 89.808%, 0.14 kW·h/g Cr(VI) removed, and $0.728/m^3^, respectively.

BBD is another optimization technique used to evaluate the effects and interactions of process variables. Shen et al. [24] employed this technique for Cr(VI) removal from K_2_Cr_2_O_7_ aqueous solution by electrocoagulation with an iron electrode. Under the optimum conditions (pH_o_: 5.48; electrode distance: 2.51 cm, J: 87.55 mA/cm^2^, and t: 25.6 min for Cr(VI) of 50 mg/L), 99.34% Cr(VI) removal efficiency was achieved. Yadav and Khandegar [72] also applied BBD to investigate the effects of voltage (5, 10 and 15 V), electrolysis time (20, 30, and 40 min) and pH (3, 5, and 7) and to optimize these parameters for Cr(VI) removal from K_2_Cr_2_O_7_ aqueous solution by electrocoagulation using either an iron or an aluminium electrode. Their predicted values of responses obtained using the model fitted well with their experimental data. Singh et al. [73] also employed RSM with BBD for Cr(VI) removal from K_2_Cr_2_O_7_ aqueous solution by electrocoagulation with an aluminium anode and an aluminium or graphite cathode. Their results indicated that the usage of a graphite cathode (0.194 kWh/m^3^) instead of an aluminium cathode (0.425 kWh/m^3^) significantly reduced the power consumption under the optimized conditions (2.38 mA/cm^2^, pH of 7.29, 23 min, and electrode distance 3.5 cm) for maximum Cr(VI) removal (74.07% for Al-Al and 70.83% for Al-graphite).

Kumar and Basu [74] adopted the Taguchi method (L9 orthogonal array (OA) with 3 factors in 3 levels) for the optimization of current density, initial Cr(VI) concentration, and initial pH to maximize Cr(VI) removal by electrocoagulation with aluminium electrodes. They determined the optimum working conditions as an initial Cr(VI) concentration of 15 mg/L; a current density of 49.3 mA/cm^2^; and an initial pH of 5. The experimental Cr(VI) removal efficiency (96.6%) was in excellent agreement with that of the predicted efficiency (98.4%). The operating cost corresponding to a maximum Cr(VI) removal efficiency at the optimum conditions was calculated as $10.77/m^3^.

Artificial neural network (ANN) was another technique used for the modelling of the experimental study results obtained from an electrocoagulation application performed with an iron anode and a steel cathode [44]. This model was developed using a 3-layer feed-forward backpropagation network with 4, 10, and 1 neurons in the first, second, and third layers, respectively. Since a comparison model results with experimental data gave a high correlation coefficient (R^2^ = 0.976), the usage of the model was proposed for the prediction of the residual Cr(VI) concentration in the effluent.

## 5. Reactor Design and Other Issues Related to Electrode Type

In scientific studies, glass beakers with different working volumes (0.25–5 L) or simple reactors made from different materials such as plexi-glass (P-G), poly-ethylene (P-E), and polymethylmethacrylate (PMMA) in rectangular or cylindrical form were used as electrocoagulation units. These electrocoagulators were operated either in batch or continuous mode (Table 1).

In general, the electrodes were connected using various monopolar (MP) configurations. In these studies, current density was kept constant by means of a direct current (DC) power except for a few studies. To compensate for some drawbacks of DC-EC such as passivation and large energy consumption, some authors recommended the use of alternating current (AC) and pulse current (PC) [3,64,75].

In a study [71] where electrocoagulation was performed using an iron anode and an aluminium cathode, the alternating pulse current (APC) mode was found to be more efficient than the DC mode with a lower anode over-voltage and slower anode polarization and passivity.

**Table 1 molecules-28-02411-t001:** Reactor designs, electrode types, and arrangements.

Electrocoagulator	Electrodes	A/C (ED) ^1^	Connection	Ref.
500 mL PMMA (100 × 100 × 50 mm)	Fe; 2 sections (50 cm^2^)	1/1 (NA ^2^)	DC or PC	[3]
2 L glass beaker (d:11 cm)	Fe; 4 rod+1 circular plate	4/1(0.87 cm)	DC	[19]
4 L P-G cylindrical (d:10 cm)	MS ^3^ rod anode + HC Fe MC ^4^	NA	DC	[20]
2 L P-E rectangular	SS ^5^; 6 rod	3/3 (2 mm)	DC	[21]
1 L cylindrical	Fe; 2 flat plate (80 cm^2^)	1/1 (18 mm)	DC	[23]
500 mL glass beaker	Fe; 2 flat plate	1/1(NA)	DC/BP	[24]
2.5 L	Fe; 2 rod (16 cm^2^)	1/1 (25 mm)	DC	[25]
P-G rectangular	2 SS	1/1 (15 cm)	DC	[27]
P-G rectangular (L:109 w:74 h:208 mm)	Fe or Al; 6 flat plates	3/3 (10 mm)	MP/DC	[28]
2 L beaker	MS; 4 flat plates	NA (5 mm)	DC/BP	[30]
NA	Fe or Al sheet (7 × 7.7 cm) Pt Ti ^6^ anode/Fe or Al cathode	1/1 (4 cm)	DC	[31]
2 L beaker	Fe anode/steel cathode	1/1 (1.4 cm)	DC	[39]
N.A.	Fe or Al anode/steel cathode	1/1 (NA)	DC	[44]
P-G cylindrical (d:15 h:13 cm)	2, 4, 6	NA (1 cm)		[45]
5 L glass	Fe, Al or SS-2 Plate	1/1; NA	MP/DC	[46]
P-G cylindrical (d:14 h:24 cm)	Fe- 2 concentric	1/1 (0.5 cm)	MP/DC	[48]
250 mL glass beaker	2 plate electrode (1 cm^2^)	1 × 1(0.5 cm)	DC	[50]
Glass beaker	2 steel slice (2 × 2 cm)	1/1 (2 cm)	DC	[57]
2.5 L	Fe-2 iron (20 cm^2^)	1/1 (2 cm)	DC	[59]
3 L P-G cylindrical	Al/Al ^7^ (210.5 cm^2^)	1/1(2 cm)	DC	[60]
P-G rectangular (24 × 17 × 18 cm)-St ^8^	Al, 4 plates	2/2 (1.5 cm)	DC	[61]
1.5 L rectangular (L:15 w:10 h:12 cm)	Al (12.5 × 8 × 1 cm)	NA (2.5 cm)	DC	[62]
P-G cylindrical (d:15 h:25 cm)	Al, 6 (d: 2 cm)	NA (1 cm)	DC	[63]
1 L reactor	Al, 4 (150 × 60 × 2 mm)	NA (1 cm)	DC/DP	[64]
5 L P-G cylindrical reactor	Al, (d:15 cm; 126 cm^2^)	NA (3 cm)	DC	[65]
1 L borosilicate glass reactor	Al, 2 plates (30.74 cm^2^	1/1 (22 mm)	DC	[66]
400 mL glass beaker	Fe or Al sheets	1/1 (1.5 cm)	DC	[72]
Acrylic rectangular (L: 7 w: 4 h: 30 cm)	2 Al sheet(100 cm^2^)	1/1 (15 mm)	DC	[70]
2 L P-G cylindrical reactor	2 Al (81.056 cm^2^)	1/1 (2 cm)	DC	[74]
7 L P-G rectangular (18 × 18 × 30 cm)	Al/Al or graphite (450 cm^2^)	1/1 (2–4 cm)	DC	[73]
0.7 L reactor	Fe/Al (36 cm^2^)	1/1 (1.5 cm)	DC or AC	[75]
2651 mL P-G (d:150 h:150 mm)	Fe/Fe or Al/Al (63 cm^2^)	1/1 (NA)	DC	[76]
1 L Pyrex reactor	Fe/SS	NA	DC	[77]
P-G rectangular (6.45 × 9.95 × 11.2 cm)	6 Fe, Al, SS combinations ^9^	3/3 (6 mm)	MP/DC	[78]
1L P-P beaker	2 Fe/Fe rod	1/1 (NA)	NA	[79,80]
P-G cylindrical (d: 9 cm; h:13 cm)	Fe/Fe (85 cm^2^)	1/1 (NA)	DC	[81]

^1^ A/C; ED: anode/cathode; electrode distance; ^2^ NA: not available; ^3^ MS: mild steel; ^4^ HC Fe mesh C: hollow cylindrical Fe mesh cathode; ^5^ SS: stainless steel; ^6^ Pt Ti: platinized titanium; ^7^ vertically rotating cylindrical aluminium electrodes; ^8^ St: separate settling tank-continuous system; ^9^ total area of 197.91 cm^2^; DC = direct current; DP = dipolar; MP = monopolar; PC = pulse current; NA= not available.

The performance of positive single pulse current (PSPC) and alternating pulse current (APC) was also compared with that of DC in the study of Zhou et al. [3] where electrocoagulation utilized iron anode and cathode. For all cases, Cr(VI) removal efficiencies were over 99%. Among all types of currents tested, PSPC consumed the least electrical energy and APC produced the least amount of sludge. The energy consumptions of DC, PSPC, and APC were calculated at 3.8 × 10^−3^, 4.0 × 10^−4^, and 7.6 × 10^−4^ kWh/mgCr(VI), and the amounts of dry sludge produced were determined as 1.8612, 1.3024, and 1.1246 g/L, respectively. The flocs produced by APC had a larger surface to adsorb Cr(VI) and formed larger particles, as compared with DC [3].

As aforementioned, iron and aluminium were commonly used as sacrificed electrode materials. In some studies, the effect of the electrode material on Cr(VI) removal efficiency was investigated using several combinations of these materials as the anode and cathode [26,77,78]. Different combinations of iron, stainless steel, aluminium, and copper as the anode or cathode material were found to be more efficient in Cr(VI) removal or as an energy-efficient combination [26,29,43,44,78]. A reactor with a centrifugal electrode with an aluminium anode and cathode was also designed so as to improve the treatment performance [61]. Compared with stationary electrodes, Cr(VI) removal efficiency significantly improved and anode passivation was reduced by the usage of the centrifugal electrode. Another cell design consisting of two concentric vertical cylindrical Fe-electrodes was also developed by Zewail et al. [48]. The outer electrode was a cylindrical solid Fe-cathode supported in the vessel wall and the inner electrode was an expanded cylindrical Fe-anode. Such a design reduced power consumption (1.09 and 2.299 KWh/m^3^ for Cr(III) and Cr(VI) removal, respectively). The effect of electrode position on Cr(VI) removal efficiency by electrocoagulation with mild steel electrodes was also explored [49]. Vertically orientated electrodes exhibited higher Cr(VI) removal performance than horizontally orientated electrodes.

The effect of distance between the anode and cathode on the process performance was also investigated. Larger distances between the anode and cathode yielded lower Cr(VI) removal efficiencies due to an increase in the resistance offered by the cell [39,46]. 

## 6. Sludge Characteristics

Almost all electrocoagulation experiments reviewed in this paper were realized under gentle mixing conditions using either a magnetic stirrer or proper apparatus due to the positive effect of the mixing on the flocs formation [39]. To prevent a concentration gradient, vertically rotating cylindrical aluminium electrodes were also designed and used for Cr(VI) removal by electrocoagulation [60]. The 3D response surface plots obtained from RSM with CCD were used to determine the interaction between rotation speed and initial pH and current or initial Cr(VI) concentration. Data indicated that (i) an increase in rpm from 40 to 70 gradually improved Cr(VI) removal efficiency and (ii) increasing rpm to 100 caused excessive agitation that led to the disintegration of the flocs due to hydrodynamic shear. These smaller flocs had excessively poor settling characteristics together with low Cr(VI) removal efficiency.

Hamdan and El-Nas [76] characterized the sludge generated during electrocoagulation using iron electrode. In their study, the surface morphology of the sludge was evaluated using field emission scanning electron microscope (FE-SEM); chemical composition and the elemental composition of the sludge were determined using energy dispersive X-ray spectroscopy (EDS) and X-ray fluorescence (XRF), respectively. The analyses confirmed the formation and precipitation of Fe(OH)_3_ and Cr(OH)_3_ as solids. The sludge consisted of iron oxide with 84.1 mass % in the form of Fe_2_O_3_, which resulted from the thermal decomposition of Fe(OH)_3_. Similar data were reported for aluminium electrodes by Kumar and Basu [74]. According to their XRF results, Al_2_O_3_ (81.5%), P_2_O_5_ (13%), and Cr_2_O_3_ (3.12%), as the significant (by mass percent) element oxides, were present in the sludge sample. The presence of aluminium oxide in sludge as Al_2_O_3_ indicated a thermal decomposition of Al(OH)_3_.

The presence of amorphous and flak-shaped aggregates was evident in FE-SEM images for the sludge produced during electrocoagulation with iron electrodes [20,76]. Sludge (flocs) characteristics for Al anode and either Al or graphite cathode were examined in the study of Singh et al. [73]. While the graphite cathode arrangement generated highly dense flocs with a small size and porous structure, the aluminium cathode arrangement produced less porous and larger size flocs in granular form.

Sludge characteristics in terms of handling and dewatering of sludge generated from electrocoagulation using stainless steel (SS) electrodes were also explored by Ölmez [21]. The specific resistance to filtration (SRF) test and the leaching test (EN 12457-4) were conducted. The dryness of the sludge was found as 3% and 18% before and after the SRF tests. The SRF of the EC sludge varied between 7.8 × 10^12^ and 10.1 × 10^12^ m/kg exhibiting good dewaterability characteristics for her optimum conditions. The leaching test result (0.5 mg/L) confirmed the non-hazardous nature of the generated sludge.

## 7. Kinetics Analysis

Cr(VI) reduction rate is dependent on several operation parameters such as initial Cr(VI) concentration, current density, initial pH, electrolyte concentration, and electrode distance. Until now, kinetic analysis has been made to evaluate the effect of these operation parameters on the electrocoagulation process performance in a few scientific types of research (Table 2). In general, the pseudo-first-order kinetic model (Equation (25)) was used to describe the removal rate of Cr(VI) by electrocoagulation due to its good fit to experimental data.
(25)$$\ln\frac{{\left[\mathrm{Cr}(\mathrm{VI})\right]}_{t}]}{{\left[\mathrm{Cr}(\mathrm{VI})\right]}_{o}]}=-k_{1}\text{×}\;t$$
where Cr(VI)_o_ and Cr(VI)_t_ are the molar Cr(VI) concentrations (in mol/L) at the initial and at a certain electrolysis time, respectively. T is the electrolysis time, and k_1_ is the pseudo first-order reaction rate constant.

The pseudo-first-order model was also rearranged by Khan et al. [20] as follows:ln(q_e_ − q) = ln(q_e_) − k_1_ × t(26)
where q_e_ is the adsorption capacity at the equilibrium (in mg/g), q is the ratio of the amount of adsorbate adsorbed at t, to the amount of adsorbent present. Furthermore, Yu et al. [65] analysed their data obtained for different electrode rotating speeds (ERSs) using the variable order kinetic (VOK) model. The model was derived as a combination of the pseudo-first-order kinetics and Langmuir adsorption isotherm. In this model, the amount of aluminium liberation from the anodes during electrocoagulation is incorporated into the Cr(VI) removal rate equation as follows:(27)$$-\frac{{\mathrm{dC}}_{t}}{\mathrm{dt}}={\;\varepsilon }_{\mathrm{Al}}\times \varepsilon_{C}\times \frac{I}{z\times F\times V}\times \frac{q_{\max}\times {\;k}_{L\;}\times {\;C}_{t}}{1+k_{L}\times {\;C}_{t}}$$
where $\varepsilon_{Al}$ and $\varepsilon_{C}$ are the efficiencies (%) of Al(OH)_3_ formation and the current efficiency, respectively. In the formula, I is the applied current (A); z is the valence of the metal of the anode (=3 for aluminium); F is the Faraday constant (96,500 C); V is the working volume of the water (7 L in the study of Yu et al. [65]); q_max_ represents maximum adsorption capacity; k_L_ is the Langmuir constant (6000 L mol^−1^ in the study of Yu et al. [65]); and C_t_ is the residual Cr(VI) molar concentration (mol/L). A quantitative model for the dynamics of Cr(VI) reduction in electrocoagulation with iron electrodes was proposed based on studies of Cr(VI)/Fe(II) and Fe(II)/O_2_ systems by Pan et al. [79]. In their model, the rates of change of Cr(VI) and Fe(II) during electrocoagulation were expressed as
−d[Cr(VI)]/dt= k_homo_[Cr(VI)]_diss_[Fe(II)]_diss_ + k’_hetero_[Fe(III)]_s_ [Cr(VI)]_diss_[Fe(II)]_diss_(28)
−d[Fe(II)]/dt= 3k_homo_[Cr(VI)]_diss_[Fe(II)]_diss_ +3k’_hetero_[Fe(III)]_s_ [Cr(VI)]_diss_[Fe(II)]_diss_ − k_2_ + k_O2_[Fe(II)]_diss_(29)

In Equations (28) and (29), k_homo_ is the homogeneous rate constant for the reduction of Cr(VI) by Fe(II) (1/Ms); k’_hetero_ is the heterogeneous rate constant for Cr(VI) reduction by adsorbed Fe(II) (1/(M^2^ s); k_2_ is the Fe(II) generation rate in EC (M/s) and was calculated from Faraday’s law as 1.92 × 10^−7^ for 37 mA/cm^2^ (M/s); and k_O2_ is the Fe(II) oxidation rate by O_2_ (1/s) and was derived from [82]. In the studies of Pan et al. [79,80], k_homo_ was determined as 35, 811, 51.5 × 10^3^, and 4.85 × 10^6^ for pH 6, 7, 8, and 9. K’_hetero_ was derived as 1.1 × 10^7^ for pH 6. It was reported that (i) the total Cr(VI) predicted by the model agreed well with the measured dissolved Cr(VI) at both oxic and anoxic conditions; (ii) as the generated Fe(II) from the anode was immediately oxidized by Cr(VI), the heterogeneous reaction did not occur at pH ≥ 7.

The limited data on electrocoagulation with iron electrode indicated that Cr(VI) removal rates (i) fitted pseudo-first-order kinetics very well (R^2^ > 95), and (ii) the rate of Cr(VI) removal decreased with increasing initial Cr(VI) concentration [20,46]. This behaviour was generally explained by a decrease in the ratio of Fe(OH)_3_ to the initial Cr(VI) concentration in the solution. Kinetic analysis performed for electrocoagulation with copper electrodes showed that (i) Cr(VI) removal followed the first-order kinetic model with respect to initial Cr(VI) concentration (R^2^ = 0.868), current density (R^2^ = 0.89), and electrode distance (R^2^ = 0.962), and (ii) the first order kinetic rate constants can be calculated using empirical equations given in Table 2. The VOK model results were reported as being in good agreement with the experimental data [65]. The model results were interpreted as the maximum adsorption capacity negatively affected by increasing the rotating speed due to breakage of flocs as the fastest rotating speed resulted in the lowest q_max_.

## 8. Electrocoagulation Application to Industrial Effluents

As seen in Table 3, a limited number of studies have dealt with Cr(VI) removal from industrial effluents by electrocoagulation utilizing sacrificed electrodes. These studies have proved that the electrocoagulation process exhibits a great potential for both the reduction of Cr(VI) to Cr(III) and the removal of Cr(III) as Cr(OH)_3_ together with other pollutants such as organic matter, oil-grease, and metals. When properly implemented, almost complete chromium removal efficiencies ensuring discharge standards could be achieved by the application of electrocoagulation. In these studies, it was emphasized that the operating cost comprised of electricity and sludge handling and disposal is one important concern for the application of electrocoagulation to industrial effluents. Some researchers recommended the use of statistical techniques for the minimization of sludge production and the operating cost [21,27]. Within this context, the optimization of operating conditions corresponding to maximum chromium removal is reported as an economically applicable approach to encourage the usage of electrocoagulation in practice. All data presented in Table 3 were produced using bench-scale reactors operated under laboratory conditions. To the best of our knowledge, there is no publication reporting the efficacy of pilot-plant or full-scale electrocoagulation reactors on Cr(VI) removal from industrial effluents.

## 9. Electroreduction by Redox Electrodes and Dimensionally Stable Electrodes

Redox electrodes which do not release matter, but exchange electrons, have long been used for the determination of Cr(VI) and chromium: gold-sputtered plastic electrode (Au-PET) [4], gold plated poly ethylene terephthalate (PET) [84], boron-doped diamond [85], gold, glassy and boron-doped diamond [86] carbon, gold, bismuth, and mercury [87]. Both redox electrodes and dimensionally stable electrodes have also been used for chromium reduction. Gold electrodes and carbon electrodes [15], titanium and graphite electrodes [88], and gold nano-particles decorated TiO_2_ [15] were utilized for this purpose. The use of conducting polymers as electrode covers is based on their ability to be oxidized while the ions in the solution, Cr(VI), are reducing. Then, the polymer can be reduced by cathodic potential application. Wei et al. [89] were one of the first authors who used polyprrole for Cr(VI) reduction in 1993. Tian et al. [90] applied polyprrole on steel, while Conroy [91] used polyprrole on aluminium for Cr(VI) reduction. Ruotolo and Gubulin [92] used a thin film of polyaniline instead of polyprrole and obtained current efficiencies reaching 100%. The Cr(VI) reduction efficiencies obtained in these works can be assumed satisfactory and the efficiency was mostly found to be independent of applied potential as was the film thickness applied onto the base metals. Marghaki et al.[17] used immobilized Fe_3_O_4_ as the magnetic nanoparticles on microbial cellulose, which was then modified and conducted by polyprrole together with an SS cathode or as both anode and cathode. The mechanism of the developed anode was explained as the adsorption of Cr(VI). Cathodic Cr(VI) reduction was also claimed to occur. Using this method, 50 mg/L Cr(VI) removed over 99% efficiency with 0.62 kWh/m^3^ energy consumption and a low sludge production of 0.018 kg/m^3^.

The use of redox electrodes and dimensionally stable electrodes, particularly polymer-type electrodes, may have a potential for use in chromium reduction. Their main advantage is the reduction of the amount of sludge produced to a minimum, which means being limited to the amount of chromium hydroxide. Their regenerative usage is another advantage. However, the above data outlined on their efficiency, power requirement, and operation practices seem to be inadequate for making decisions and comparisons with conventional methods. Therefore, as new electrode materials are developed, future studies are expected to shed more light on the subject.

## 10. Conclusions

Electrocoagulation is a flexible process and is applicable to diversified wastewater as a treatment method. The data collected in the literature verify its ability to deal with Cr(VI) removal from wastewaters. Chromium removal is realized in two consecutive steps: Cr(VI) reduction and Cr(III) precipitation. The advantage of electrocoagulation is the combination of these two processes. The reduction process needs an acid medium to occur at an acceptable rate; the precipitation, however, requires at least a near-neutral medium to effectively remove Cr(III) as Cr(OH)_3_. The combination of these two processes is carried out in electrocoagulation through the shift of pH upwards as the process runs. This needs a careful design and operation of electrocoagulation, particularly as an iron electrode is used as the anode. On the other hand, the electrocoagulation process itself inherently involves several operating parameters that further complicate the operation. For the use of iron electrodes, the following points seem to be of importance. Acidic initial pH is, as commonly expressed, favourable; however, it seems to be related to initial chromium concentration as well as the final pH reached at the end of the process. High initial chromium concentrations tend to reduce the process efficiency. Current density is the main parameter determining the cost of the process. High current densities enhance the chromium removal mostly through an increasing Fe(II) production. Chloride stands as the best-supporting electrolyte. Aluminium electrodes behave in a similar way to that of iron electrodes in terms of the main operation parameters such as initial chromium concentration and current density. However, the reduction process occurs at the cathode instead of the anode. Among the other electrodes, the use of redox and dimensionally stable electrodes seems to have potential due to a significant reduction in residual sludge; however, the cost of the process may be limiting. For all electrodes and reactor configurations being used or to be developed, further research is needed to provide sound bases for the design and operation of electrocoagulation for Cr(VI) treatment.

## Figures and Tables

**Figure 1 molecules-28-02411-f001:**
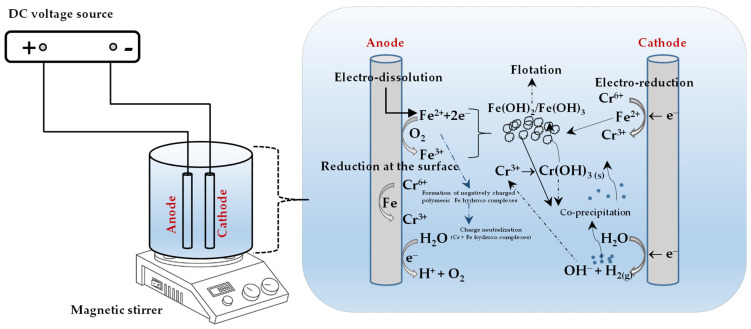
Possible Cr(VI) removal mechanisms for EC with iron-based electrodes.

**Figure 2 molecules-28-02411-f002:**
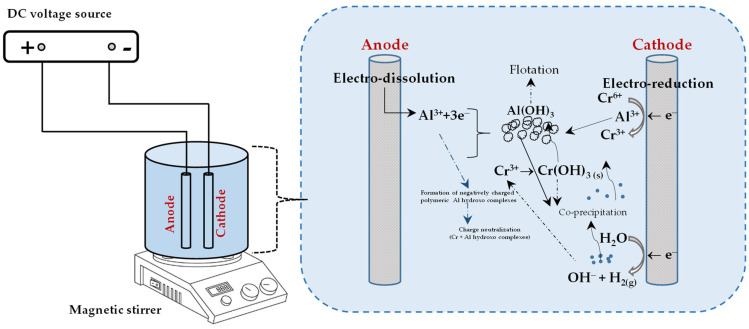
Possible Cr(VI) removal mechanisms for EC with aluminium electrodes.

**Table 2 molecules-28-02411-t002:** The Cr(VI) removal rate constants and q_m_ for the VOK model (for 60 min; R^2^ > 0.95 except those of [8]).

		Cr(VI)_o_ (mg/L)	A/C	Operation Condition	Ref
$k_{1}$	=2.32 × 10^−4^ × CD	100	Cu/Cu	pH_o_: 5.65; C: 0.1–0.3 A	[8]
$k_{1}$	=−5.14 × 10^−4^ × C_o_ + 0.109	50–150	Cu/Cu	pH_o_: 5.65; C: 0.3 A	[8]
$k_{1}$	=−0.0128 × ED + 0.083	100	Cu/Cu	pH_o_: 5.65; C: 0.3 A	[8]
$k_{1}$	0.0305	60	Fe/Fe	pH_o_: 3; J: 0.2 mA/cm^2^	[20]
$k_{1}$	0.0226	80	Fe/Fe	pH_o_: 3; J: 0.2 mA/cm^2^	[20]
$k_{1}$	0.0139	100	Fe/Fe	pH_o_: 3; J: 0.2 mA/cm^2^	[20]
k_1_	0.1271	10	Fe/Fe	pH_o_: 4; J: 43.03 A/m^2^	[46]
k_1_	0.1171	40	Fe/Fe	pH_o_: 4; J: 43.03 A/m^2^	[46]
$k_{1}$	0.0729	50	Fe/Fe	pH_o_: 4; J: 43.03 A/m^2^	[46]
$k_{1}$	0.0148	100	Fe/Fe	pH_o_: 4; J: 43.03 A/m^2^	[46]
q_max_	14.06	50	Al/Al	pH_o_: 5; J: 25 mA/cm^2^; ERS: 120 rpm	[65]
q_max_	12.78	50	Al/Al	pH_o_: 5; J: 25 mA/cm^2^; ERS: 180 rpm	[65]
q_max_	10.11	50	Al/Al	pH_o_: 5; J: 25 mA/cm^2^; ERS: 240 rpm	[65]

k_1_: in min^−1^; CD: current density (in A/m^2^); C_o_: initial Cr(VI) concentration (in mg/L); ED: electrode distance (in cm); q_max_: the maximum adsorption capacity (in mg adsorbate absorbed/g adsorbent).

**Table 3 molecules-28-02411-t003:** Electrocoagulation application to industrial effluents.

Samples	A/C	Optimum Operating Conditions	Efficiency [%]	Ref
Hard chromium coating	SS/SS	pH_o_ 1.85; 7.4 A; 70 min; NaCl: 33.6 mM; 1470 mg/L^1^	100	[21]
Chrome bathwater	SS/SS	pH_o_ 5; 68 A/m^2^; 17 min; 1500 mg/L^1^	97.5	[27]
Electroplating effluent	Fe/S	pH_o_ 6.9; 50 mA/cm^2^; 30 min; 17.1 mg/L^1^	97	[44]
Electroplating effluent	Fe/Fe	pH_o_ 4; 50 mA/cm^2^; 15 min; 889.29 mg/L^1^	100	[47]
Surface treatment effluent	Al/Al	pH_o_ 7; 8.6 A; 60 min; 10020 mg/L^1^	99.99	[62]
Batik industry	TD ^2^-Al/SS	pH_o_ 7;15 V; 4 h; 3 mg/L^1^	100	[77]
Metal plating effluent	Fe/Fe	pH_o_ 7.4; 35 mA/cm^2^; 30 min; 358 ± 2.1 mg/L^1^	98.9	[81]
Metal plating effluent	Fe/SS	pH_o_ 2.42; 50 mA/cm^2^; 30 min; 13.9 mg/L^1^	98	[83]

^1^ initial chromium concentration; ^2^ TD-Al: titanium dioxide coated aluminium; SS = stainless steel.

## Data Availability

Not applicable.

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
