# Peer review of "Hexavalent Chromium Removal from Water and Wastewaters by Electrochemical Processes: Review"

_molecules, 2023, doi:10.3390/molecules28052411_

Round 1

Reviewer 1 Report

The authors provided an interesting review on  hexavalent chromium removal from water and wastewater by chemical and electrochemical processes. The last, are the predominant techniques described in the manuscript. I think that  a revision of the  review structure is advisable. In fact, great attention and space in the review was done to Electrocoagulation (section 3).  Moreover, I suggest to add tables and figures that summarize the numerous information reported in the text in order to make it shorter and more fluent.

A deepen English revision is required.

Author Response

Answer: Thank you very much for your constructive comments. The text body has been restructured as seen in the new version. Two new figures (Figs. 1 and 2) have been included in the text. Entire text has been edited.  

Reviewer 2 Report

This review mainly focuses on the chemical and electrochemical processes of electrocoagulation, so the keyword could be out into the title and the related technology could be also reviewed.

The reclamation of the chromium waste should be also discussed.

Author Response

Answer: Thank you very much for your constructive comments. The text body has been restructured as seen in the new version of the manuscript. Title has been also revised.   

The reclamation of the chromium waste should be also discussed.

Answer: For the chromium which is a hazardous waste, reclamation and recovery is important. Our paper is a review article and we can present data or make a discussion if any subject has been dealt with in the literature. Unfortunately, to the best of our knowledge there is no publication covering the subject of reclamation. However, considering the importance of the subject we pointed out the need of such studies in the conclusions.

Author Response

The author summarized Chromium reduction by chemical method and electro coagulation

techniques in detail. This is very general review and the sequence of arrangement of topics

and content coverage are written very poor. The following points may be addressed and may

be added to strengthen the manuscript.

Answer: Thank you very much for your constructive comments.  Chemical reduction section has been removed. The text body has been restructured as seen in the new version. Abstract and Conclusions have been revised/rewritten. The additions requested to strengthen the manuscript have been made.

  1. Abstract is written in very poor form. Rewrite the abstract section.

Answer: It has been revised. Now, its capacity is at the limit of 200 words

  1. Optimization study heading is irrelevant to this review.

Answer: This heading has been changed as “4. Optimum operating conditions and interactions of process variables”  

  1. Check the equation (equation 34)

Answer: It has been used as was given in the paper of Khan et al. (Khan, S.U.; Islam, D.T.; Farooqi, I.H.; Ayub, S.; Basheer, F. Hexavalent chromium removal in an electrocoagulation column reactor: Process optimization using CCD, adsorption kinetics and pH modulated sludge formation. Process Safety and Environmental Protection 2019, 122, 118-130, doi:10.1016/j.psep.2018.11.024.).

  1. Table 3, qmax unit is not given.

Answer: It has been given as a footnote of Table 3.

  1. Author not discusses the electrode potential of different electrodes in the review.

Answer: Electrode potential is not an informative parameter for the EC. None of the articles we have studied through writing this paper, which are over 100, made use of electrode potentials as a parameter. However, we have provided standard electrode potentials for both iron and aluminum in the text.

  1. The author can list out the advantage and disadvantages of each process in details.

A paragraph discussing the comparison of iron and aluminum electrodes has been added in the text (at the end of aluminum electrodes section)

  1. Rewrite the conclusion part. It is unclear.

Answer: It has been rewritten.

  1. Reactor configuration can be explained with diagram in detail.

Answer: Figures 1 and 2 have been included.

Reviewer 4 Report

- The article is well organized and interesting. It should be supported by a few strong sentences about the novelty of the work.

- A graphical summary supporting the general decision of the study could be attached.

- The reactions under heading "2. Chemical Hexavalent Chromium Reduction" need to be controlled.

-The acid compound on line 96 should be checked.

-The pH sentences on line 273 should be checked.

- The Cr(VII) sentences on line 286 should be checked.

-The units of the parameters in table 2 and table 3 should only be given once in the relevant column.

-In addition, Tables 2, 3 and 4 can be enriched with the following articles. The following references must be given for the development of the introduction or Results and discussion sections of the article;

- Oden, M. K. (2020). Treatment of CNC industry wastewater by electrocoagulation technology: an application through response surface methodology. International Journal of Environmental Analytical Chemistry, 100(1), 1-19.

-Oden, M. K., & Sari-Erkan, H. (2018). Treatment of metal plating wastewater using iron electrode by electrocoagulation process: Optimization and process performance. Process Safety and Environmental Protection, 119, 207-217.

- Evliyaogullari, N. E., Oden, M. K., & Kucukcongar, S. (2017). The removal of chromium from aqueous solutions using an industrial waste material. International Journal of Ecosystems and Ecology Science (IJEES), 7(4), 671-676.

Author Response

Reviewer 4.

The article is well organized and interesting. It should be supported by a few strong sentences about the novelty of the work.

We have added new sentences emphasizing the important outcomes of the work in the conclusions.

- A graphical summary supporting the general decision of the study could be attached.

Answer: Thank you very much for your comments. We should inform you that the editor made her/his decision on February 15, 2023, before submitting your review report (February 19, 2023). We saw your review report while submitting our revised manuscript. Therefore, we could have not drawn a graphical abstract. Additionally, the graphical abstract is not requested by the journal. But, we have included 2 new figures to enrich the manuscript.

- The reactions under heading "2. Chemical Hexavalent Chromium Reduction" need to be controlled.

Answer: Section 2 has been removed in the text as requested by other reviewer. The text has been restructured.

-The acid compound on line 96 should be checked.

Answer: Section 2 has been removed in the text as requested by other reviewer.

-The pH sentences on line 273 should be checked.

Answer: It has been changed.

- The Cr(VII) sentences on line 286 should be checked.

Answer: It has been changed.

-The units of the parameters in table 2 and table 3 should only be given once in the relevant column.

Answer: It is difficult to do this as some values were reported in different unit.

-In addition, Tables 2, 3 and 4 can be enriched with the following articles. The following references must be given for the development of the introduction or Results and discussion sections of the article;

- Oden, M. K. (2020). Treatment of CNC industry wastewater by electrocoagulation technology: an application through response surface methodology. International Journal of Environmental Analytical Chemistry, 100(1), 1-19.

Answer: As It is not directly related to the subject of hexavalent chromium removal by EC, it has not been used.

-Oden, M. K., & Sari-Erkan, H. (2018). Treatment of metal plating wastewater using iron electrode by electrocoagulation process: Optimization and process performance. Process Safety and Environmental Protection, 119, 207-217.

Answer: It has been included in Table 2 and 4 as Ref. 81

- Evliyaogullari, N. E., Oden, M. K., & Kucukcongar, S. (2017). The removal of chromium from aqueous solutions using an industrial waste material. International Journal of Ecosystems and Ecology Science (IJEES), 7(4), 671-676.

Answer: It has been cited as Ref. 11

Round 2

Reviewer 1 Report

The authors have done a relevant work in revising the entire review manuscript according to the suggestions and the comments provided by all the reviewers. The structure of the manuscript has been improved and it sounds more clear than the initial version. However, the current, revised active form of the manuscript is very complicated to be easily read and  assessed; I suggest to the authors to provide a file with the accepted revisions in order to allow a quicker reading and comprehension of their review paper.